# CDA: Contrastive-adversarial Domain Adaptation

## Abstract

Recent advances in unsupervised domain adaptation (UDA) reveal that adversarial learning on deep neural networks can learn domain invariant features to reduce the shift between source and target domains. While such adversarial approaches achieve domain-level alignment, they ignore the class (label) shift. When class-conditional data distributions significantly differ between the source and target domain, it can generate ambiguous features near class boundaries that are more likely to be misclassified. In this work, we propose a two-stage model for UDA called Contrastive-adversarial Domain Adaptation (CDA). While the adversarial component facilitates domain-level alignment, two-stage contrastive learning exploits class information to achieve higher intra-class compactness across domains resulting in well-separated decision boundaries. Furthermore, the proposed contrastive framework is designed as a plug-and-play module that can be easily embedded with existing adversarial methods for domain adaptation. We conduct experiments on two widely used benchmark datasets for domain adaptation, namely, Office-31 and Digits-5, and demonstrate that CDA achieves state-of-the-art results on both datasets.

## 1   Introduction

## 2   Introduction

Deep neural networks (DNNs) have significantly improved the state-of-the-art in many machine learning problems Dong et al. (2021). When trained on large-scale labeled datasets, DNNs can learn semantically meaningful features that can be used to solve various downstream tasks such as object classification, detection, and language processing. Yosinski et al. (2014)Zhuang et al. (2020)Yadav & Ganguly (2020). However, DNNs need to be qualified with caveats Belkin et al. (2019) - they are understood to be brittle and tend to generalize poorly to new datasets Neyshabur et al. (2017)Wilson & Izmailov (2020). Even a small shift compared to the training data can cause the deep network to make spurious predictions on the target domain. This phenomenon is known as domain shift You et al. (2019)Ben-David et al. (2010), where the marginal probability distribution of the underlying data changes across different datasets or domains. A typical solution is to fine-tune a model trained on a sufficiently labeled dataset by leveraging the limited number of labeled samples from the target dataset Chu et al. (2016)Long et al. (2017). However, in real-world problems, it might be expensive, or in some instances impossible Singla et al. (2019), to collect sufficient labeled data in the intended (target) domain leaving the fine-tuning or *transferring* process challenging to execute.

Learning a model that reduces the dataset shift between training and testing distribution is known as domain adaptation Ben-David et al. (2006). When no labeled data is available in the target domain, it is called unsupervised domain adaptation (UDA) Ganin & Lempitsky (2015)Wilson & Cook (2020), which is the focus of this work. While the earliest domain adaptation methods worked with fixed feature representations, recent advances in deep domain adaptation (DDA) embed domain adaptation modules within deep learning architectures. Thus, domain adaptation and feature learning are achieved simultaneously (end-to-end) in a single training process. One of the most well-known approaches to DDA is the use of adversarial learning for reducing the discrepancy between the source and target domain Ganin et al. (2016)Tzeng et al. (2017)Pei et al. (2018)Long et al. (2018). Adversarial domain adaptation (ADA) approaches domain adaptation as a minimax game similar to how Generative Adversarial Networks (GANs) Creswell et al. (2018) work. An

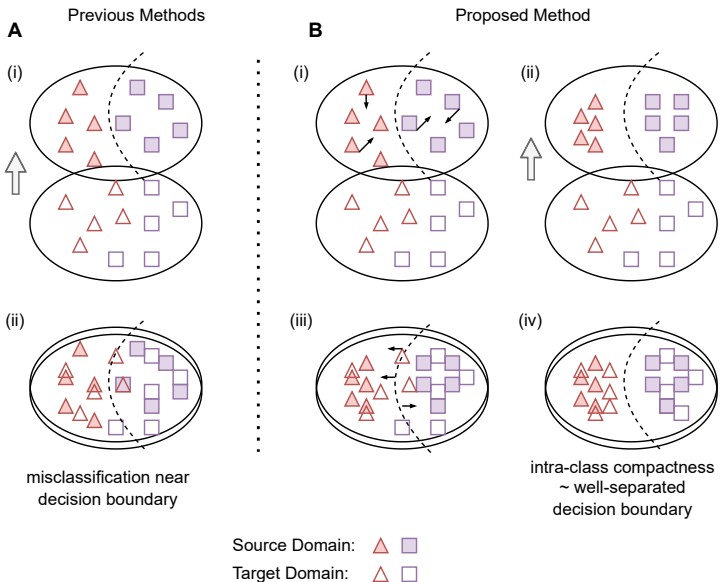

Figure 1: Illustration of the improvements proposed by CDA for unsupervised domain adaptation (UDA).(A) Existing adversarial methods for UDA align the source and target domain only at the domain level ignoring class boundaries. (B) In comparison, CDA achieves both domain and class-level alignment in a multi-step training regime. In step 1, CDA performs supervised contrastive learning on the labeled source domain, resulting in better intra-class compactness and well-separated decision boundaries for the target domain to align. In the next step, adversarial learning leads to domain-level alignment, while cross-domain contrastive learning pulls target samples to align with similar samples from the source domain and pushes away dissimilar clusters.

auxiliary domain discriminator is trained to distinguish latent feature embeddings from source and target domains. At the same time, a deep neural network learns feature representations that are indistinguishable by the domain discriminator. In other words, the deep network, comprising a generator and a dense head, and the domain discriminator try to fool each other, resulting in latent features that cannot be distinguished by which domain they come from. Although ADA achieves domain-level alignment, it fails to capture the multimodal structure within a specific domain's data distribution Wang et al. (2020)Zhang et al. (2019). Even if a domain discriminator is fully confused, there is no guarantee for class-level alignment. In scenarios where class-conditional distributions across domains are significantly different, ADA can generate ambiguous features near class boundaries that are more likely to be misclassified (see Figure 1) Chen et al. (2019a). Some of the recent works have tried to tackle the problem of class-level alignment via training separate domain discriminators Pei et al. (2018) Wang et al. (2019); however, it gives rise to convergence issues amidst a lack of equilibrium guarantee. Other works directly encode class information in the domain adaptation module Long et al. (2018)Chen et al. (2019b).

In this work, we propose a novel two-stage domain adaptation mechanism called Contrastive-adversarial Domain Adaptation (CDA). CDA leverages the mechanism of contrastive learning Le-Khac et al. (2020)Purushwalkam & Gupta (2020) for achieving class-level alignment in tandem with adversarial learning which focuses on domain-level alignment. The idea of contrastive learning is to learn an embedding space where similar data samples - and corresponding features - lie close to each other while dissimilar samples are pushed away. Although contrastive learning has been most successfully used in self-supervised learning Chen et al. (2020)Grill et al. (2020)Caron et al. (2021) tasks, the underlying idea can be exploited to solve domain adaptation. The contrastive module improves intra-class compactness (stage-I) and class-conditioned alignment (stage-II), while ADA focuses on the overall domain-level alignment. The expected outcome is a more

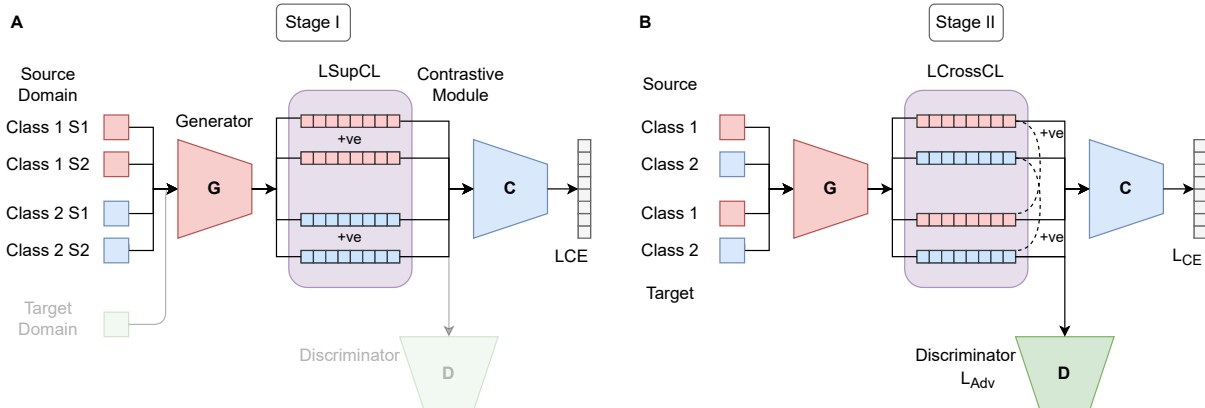

Figure 2: An overview of the two-stage CDA framework. In stage-I (A), we perform supervised contrastive learning (CL) using the labeled source dataset. The motivation is to achieve better intra-class compactness and well-separated decision boundaries to make class-level alignment in stage-II (B) easier to perform. Stage-II is where the actual domain adaptation (DA) occurs using a combination of adversarial and cross-domain contrastive loss. The overall CDA objective function comprises multiple losses that are optimized in tandem to achieve DA. For a detailed explanation, see section 3. (figure best viewed in color).

tightly coupled domain alignment that is class-aware. We conduct experiments on two benchmark datasets for UDA (Office-31 and Digits-5) to demonstrate that CDA achieves state-of-the-art results.

## 2.1 Contributions

The key contributions of this work can be summarized as follows:

- We propose a novel two-stage deep domain adaptation method (CDA) that combines contrastive and adversarial approaches for unsupervised domain adaptation (UDA).

- Experiments show the efficacy of our proposed methods by achieving state-of-the-art results on well-known benchmark datasets for UDA.

- The proposed contrastive module can be easily embedded within existing adversarial domain adaptation methods for improved performance.

## 3 Related Work

### 3.1 Unsupervised Domain Adaptation (UDA)

The central idea of UDA is to learn domain-invariant feature representations. While the earliest (shallow) approaches worked with fixed features, the current methods combine the expressiveness of deep neural networks with domains adaptation for end-to-end learning Ganin & Lempitsky (2015)Long et al. (2017)Chen et al. (2019a). There is extensive literature on deep domain adaptation methods ranging from moment matching to more recent adversarial approaches. Both approaches aim to minimize the discrepancy between the source and target domain. While moment matching methods explicitly minimize the difference using a loss function such as Maximum Mean Discrepancy (MMD) Long et al. (2015)Long et al. (2017), adversarial methods seek to reduce the discrepancy using an adversarial objective which pits two networks against each other - a generator and a discriminator. For domain adaptation, the generator's goal is to produce latent features the domain discriminator cannot classify correctly. Doing so generates domain-invariant feature representation, i.e., the target domain gets aligned with the source domain. A common criticism

of the earliest ADA methods was that they only result in domain-level alignment and ignore class-specific distributions. Recent works have built on the seminal work of Ganin et al. Ganin & Lempitsky (2015) in the context of ADA - they attempt to incorporate class-level information in the model for achieving a more tightly-coupled alignment across domains Long et al. (2018)Chen et al. (2019b)Pei et al. (2018).

## 3.2 Contrastive Learning

Contrastive learning (CL) has achieved state-of-the-art results in self-supervised representation learning Chen et al. (2020)Grill et al. (2020). The goal of CL is to learn a model where feature representations of similar samples lie close to each other in the latent space, and dissimilar samples lie further apart. In the absence of labels, an augmented version corresponding to a sample is generated to create a positive (similar) pair. The other samples in the training minibatch become negative pairs. Entropy-based loss functions that simultaneously maximize the similarity of positive pairs and minimize the similarity of negative pairs are used. Recent works Caron et al. (2021) have shown how contrastive learning can learn semantically meaningful feature representations that can be used to solve various downstream tasks, and can even outperform supervised tasks solved in supervised settings Caron et al. (2020).

## 3.3 Contrastive Learning for UDA

Recent works have applied the core principle of CL to domain adaptation tasks. Carlucci et al. Carlucci et al. (2019) used a pretext task (solving a jigsaw puzzle) for self-supervision to solve domain adaptation. Kim et al. Kim et al. (2021) proposed cross-domain self-supervised learning and extended by Yue et al.Yue et al. (2021) to align cluster-based class prototypes across domains for few-shot learning. Singh et al. Singh (2021) used CL with strongly augmented pairs to reduce the intra-domain discrepancy. Picking the appropriate augmentations for CL is heuristic and may not generalize to other datasets with the same model. We avoid data augmentation using a two-stage CL approach. To the best of our knowledge, this is the first work that systematically integrates CL with adversarial methods for the problem of unsupervised domain adaptation.

# 4 Contrastive-Adversarial Domain Adaptation

## 4.1 Problem Formulation

In UDA, we aim to transfer a model learned on a labeled source domain to an unlabeled target domain. We assume that the marginal probability distributions of the two domains are not equal, i.e., $P(\mathcal{X}_s) \neq P(\mathcal{X}_t)$. We are given a labeled source dataset $D_s = (\mathcal{X}_s, \mathcal{Y}_s) = \{(x_s^i, y_s^i)\}_{i=1}^{n_s}$ and an unlabeled dataset in the target domain $D_t = \mathcal{X}_t = \{x_t^i\}_{i=1}^{n_t}$ with $n_s$ and $n_t$ samples, respectively. Both $\{x_s^i\}$ and $\{x_t^i\}$ belong to the same set of $N$ classes with $P(\mathcal{X}_s) \neq P(\mathcal{X}_t)$. The goal is to predict labels for test samples in the target domain using the model $(\mathcal{G}, \mathcal{C}) : \mathcal{X}_t \to \mathcal{Y}_t$ trained on $D_s \cup D_t$. The trained model includes a feature generator $\mathcal{G} : \mathcal{X}_t \to \mathbb{R}^d$ and a classifier $\mathcal{C} : \mathbb{R}^d \to \mathbb{R}^N$, where $d$ is the dimension of the intermediate features produced by the generator.

## 4.2 Model Overview

CDA is a two-stage model with three major components - a feature generator $\mathcal{G}$, a classifier $\mathcal{C}$, and an auxiliary domain classifier $\mathcal{D}$ (Figure 2). Further, a contrastive module is spaced between $\mathcal{G}$ and $\mathcal{C}$. Broadly, there are two objectives achieved by the CDA model: 1) domain-level alignment using adversarial learning and 2) class-level alignment using contrastive learning. The following sections describe the mechanism of each objective in detail.

## 4.3 Domain-Level Adversarial Learning

Adversarial learning aims to learn domain-invariant features by training the feature generator $\mathcal{G}$ and domain discriminator $\mathcal{D}$ with competing (minimax) objectives. The adversarial component is adapted from the seminal work of Ganin et al. (DANN) Ganin & Lempitsky (2015) that originally proposed the idea. As a

---

**Algorithm 1:** **C**ontrastive-adversarial **D**omain **A**daptation

---

**Input** : labeled source dataset $D_s = \{\mathcal{X}_s, \mathcal{Y}_s\}$, unlabeled target dataset $D_t = \{\mathcal{X}_t\}$, max epochs $E$, iterations per epoch $K$, model $(\mathcal{C}, \mathcal{D}, \mathcal{G})$

**Output:** trained model $(\mathcal{G}, \mathcal{C})$

---

**for** $e = 1$ *to* $E$ **do**
    **for** $k = 1$ *to* $K$ **do**
        Sample batch $\{x_s, y_s\}$ from $D_s$ and compute $\mathcal{L}_{SupCL} + \mathcal{L}_{CE}$ using Eqn. 3
        **if** $e \geq E'$ **then**
            Sample batch $\{x_t\}$ from $D_t$ and compute $\mathcal{L}_{Adv}$ using Eqn. 1
            **if** $e \geq E''$ **then**
                $\mathcal{L}_{SupCL} = 0$
                Generate pseudo-labels $y_t$ and compute $\mathcal{L}_{CrossCL}$ using Eqn. 5
            **end**
        **end**
        Compute $\mathcal{L}_{Total}$ using Eqn. 7
        Backpropagate and update $\mathcal{C}, \mathcal{D}$ and $\mathcal{G}$
    **end**
**end**

---

first step in the zero-sum game, $\mathcal{G}$ takes the labeled source and unlabeled target domain inputs and generates feature embeddings $z_s$ and $z_t$. In the next step, $\mathcal{D}$ takes the feature embeddings and attempts to classify them as either coming from the source or target domain. The goal of $\mathcal{G}$ is to fool the discriminator such that output feature embeddings cannot be classified correctly by $\mathcal{D}$. It is achieved by training $\mathcal{D}$ and $\mathcal{G}$ with an adversarial loss $\mathcal{L}_{Adv}$ with gradient reversal (for $\mathcal{G}$). For a given source sample $\mathbf{x}_s \sim \mathcal{X}_s$ and target sample $\mathbf{x}_t \sim \mathcal{X}_t$, $\mathcal{L}_{Adv}$ can be formulated as a binary cross-entropy loss:

$$\mathcal{L}_{Adv}(\mathcal{X}_s, \mathcal{X}_t) = \sum_{\substack{\mathbf{x}_s \sim \mathcal{X}_s \\ \mathbf{x}_t \sim \mathcal{X}_t}} \left[ \log\left(\mathcal{D}\left(\mathcal{G}\left(\mathbf{x}_t\right)\right)\right) + \log\left(1 - \mathcal{D}\left(\mathcal{G}\left(\mathbf{x}_s\right)\right)\right) \right] \tag{1}$$

with the following objective,

$$\min_{\mathcal{G}} \max_{\mathcal{D}} \left(\mathcal{L}_{Adv}\right) \tag{2}$$

In other words, $\mathcal{G}$ tries to minimize $\mathcal{L}_{Adv}$ while $\mathcal{D}$ learns to maximize it. The theoretical argument is that convergence will result in domain-invariant feature embeddings. However, such an adversarial approach only results in domain-level alignment without considering the complex multi-mode class distribution present in the source and target domain. Even when the domain discriminator is fully confused, there is no guarantee the classifier can successfully discriminate target samples based on the class labels. The absence of class-level alignment results in under-transfer or negative transfer when the class-conditional distributions are significantly different across the two domains.

### 4.4 Class-Discriminative Contrastive Learning

To generate feature embeddings that are not domain-invariant but also class-discriminative across the two domains, CDA proposes a constrastive learning-based (CL) module. For clarification, the CL module is not a neural network per se. It is an intermediary component that links $\mathcal{G}$, $\mathcal{D}$, and $\mathcal{C}$ and where the proposed two-stage contrastive objective is optimized.

**Stage I:** The CL module performs supervised contrastive learning on the source domain. In every batch, samples from the same class are considered positive pairs, while samples from different classes are automat-

ically assigned as negative pairs. Training progresses by optimizing a modified InfoNCE loss Chen et al. (2020) where NCE stands for Noise-contrastive Estimation (see Eq. ). Although CL is best associated with self-supervised representation learning, recent works (Khosla et al. Khosla et al. (2020)) have shown that minimizing a contrastive loss can outperform the standard cross-entropy loss for supervised classification tasks. The idea is that clusters of samples belonging to the same class are pulled together in the embedding space while simultaneously pushing apart clusters of samples from different classes creating well-separated decision boundaries for better aligning the target domain samples in the next step. The combined objective function during stage-I is as follows:

$$\mathcal{L}_{StageI} = \mathcal{L}_{SupCL} + \mathcal{L}_{CE} \tag{3}$$

$$\mathcal{L}_{SupCL}(\mathcal{X}_s, \mathcal{Y}_s) = -\sum_{\mathbf{z}, \mathbf{z}^+ \in D_s} \log \frac{\exp(\mathbf{z}^\mathsf{T}\mathbf{z}^+/\tau)}{\exp(\mathbf{z}^\mathsf{T}\mathbf{z}^+/\tau) + \sum_{\mathbf{z}^- \in D_s} \exp(\mathbf{z}^\mathsf{T}\mathbf{z}^-/\tau)} \tag{4}$$

where, $\mathcal{L}_{CE}$ is the standard cross-entropy loss for multiclass classification. $\mathcal{L}_{SupCL}$ is the supervised contrastive loss applied to samples from the labeled source domain. The variable $\mathbf{z}_s$ denotes the $l_2$ normalized latent embedding generated by $\mathcal{G}$ corresponding to the input sample $\mathbf{x}_s$. The variable $\tau$ refers to the temperature scaling (hyperparameter) which affects how the model learns from hard negatives Chuang et al. (2020).

**Stage II:** For class-level alignment, CDA performs cross-domain contrastive learning. It is based on the understanding that samples belonging to the same class across the two domains should cluster together in the latent embedding space. Unlike supervised CL in stage-I, samples from the same class across domains are considered positive pairs, and samples from different classes become negative pairs. However, we need labels for the target domain which are not available. Some of the current methods in this space generate pseudo-labels using k-means clustering Singh (2021). Clustering on the source domain is either performed once during preprocessing or performed every few epochs during training, and target labels are assigned based on the nearest cluster centroid. We argue that both approaches are sub-optimal and propose making target label generation part of the training process itself without the need to perform clustering.

$$\mathcal{L}_{CrossCL}(\mathcal{X}_s, \mathcal{Y}_s, \mathcal{X}_t) = -\sum_{\substack{i=1 \\ \mathbf{z}_s \in D_s \\ \mathbf{z}_t \in D_t}}^{N} \log \frac{\exp(\mathbf{z}_s^{i\mathsf{T}}\mathbf{z}_t^i/\tau)}{\exp(\mathbf{z}_s^{i\mathsf{T}}\mathbf{z}_t^i/\tau) + \sum_{i \neq k=1}^{N} \exp(\mathbf{z}_s^{i\mathsf{T}}\mathbf{z}_t^k/\tau)} \tag{5}$$

where, $\mathcal{L}_{CrossCL}$ is the cross-domain contrastive loss in stage-II. $\mathbf{z}_s$ and $\mathbf{z}_t$ are the $l_2$ normalized embeddings from the source and target, respectively. The superscript $i$ and $k$ are used to identify the class labels (pseudo labels in case of target domain).

### 4.5 CDA: Overall Framework

In CDA, we take a multi-step approach to optimize multiple objective functions during training. In the first stage, we train only on the source domain for the first $E'$ epochs (hyperparameter) to ensure the model reaches a certain level of classification accuracy.

Next, we initiate the process for domain-level alignment as described above. We add $\mathcal{L}_{Adv}$ to the overall objective function using a time-varying weighting scheme lambda. Once we have achieved well-separated clustering in the source domain and some level of domain alignment, we gradually introduce the last loss function $\mathcal{L}_{CrossCL}$. The (pseudo) target labels are obtained by executing a forward pass on the model $(\mathcal{G}, \mathcal{C})$: $\mathbf{y}_t = \mathrm{argmax}(\mathcal{C}(\mathcal{G}(\mathbf{x}_t)))$. Some target samples are expected to be misclassified initially, but as the

training continues and target samples get aligned, decision boundaries will get updated accordingly, and model performance will improve with each iteration. $\mathcal{L}_{CrossCL}$ pulls same-class clusters in the two domains closer to each other and pushes different clusters further apart. Finally, we also employ a standard cross-entropy loss function $\mathcal{L}_{CE}$ during the entire training process to keep track of the classification task. The overall training objective can be formulated as follows:

$$\mathcal{L}_{Total} = \mathcal{L}_{Stage1} + \mathcal{L}_{Stage2} \tag{6}$$

$$\begin{aligned} \mathcal{L}_{Total} &= \mathcal{L}_{SupCL} + \mathcal{L}_{CE} \\ &+ \lambda * \mathcal{L}_{Adv} + \beta * \mathcal{L}_{CrossCL} \end{aligned} \tag{7}$$

where

$$\lambda = \begin{cases} 0 & \text{for epoch } 0 \leq e < E' \\ \frac{2}{1+\exp^{-\gamma p}} - 1 & \text{for epoch } e \geq E' \end{cases} \tag{8}$$

and

$$\beta = \begin{cases} 0 & \text{for epoch } e \leq E'' \\ \min(1, \alpha * \left(\frac{e-E''}{E''}\right)) & \text{for epoch } E'' < e \leq E \end{cases} \tag{9}$$

where, $E'$ and $E''$ (with $E'' \geq E'$) indicate the epochs when Stage-I ends and $\mathcal{L}_{CrossCL}$ is introduced in the objective function, respectively. At any given epoch, only one type of contrastive learning is performed, i.e. for $e \geq E''$, $\mathcal{L}_{SupCL} = 0$ (see Algorithm 1). The scaling variables $\lambda$ and $\beta$ (hyperparameters) control the rate at which $\mathcal{L}_{Adv}$ and $\mathcal{L}_{CrossCL}$ are added to the overall objective function to maintain the stability of the training process. The values of $\alpha$ and $\beta$ increase from 0 to 1.

## 5 Experiments

### 5.1 Datasets

We use two public benchmarks datasets to evaluate our method:

**Office-31** is a common UDA benchmark that contains 4,110 images from three distinct domains – Amazon (**A** with 2,817 images), DSLR (**D** with 498 images) and Webcam (**W** with 795 images). Each domain consists of 31 object classes. Our method is evaluated by performing UDA on each pair of domains, which generates 6 different tasks (Table 1).

**Digits-5** comprises a set of five datasets of digits 0-9 (MNIST, MNIST-M, USPS, SVHN, and Synthetic-Digits) most commonly used to evaluate domain adaptation models. We use four of the five datasets and generate 3 different tasks (Table 2). Both MNIST and MNIST-M contain 60,000 and 10,000 samples for training and testing respectively. SVHN is a more complex real-world image dataset with 73,257 samples for training and 26,032 samples for testing. The digits in SVHN are captured from house numbers in Google Street View images. SVHN has an additional class for the digit '10' which is ignored to match the label range of other datasets. Finally, USPS is a smaller dataset with 7,291 training and 2,007 testing samples. We use all the available training samples for each task.

DANN CDA

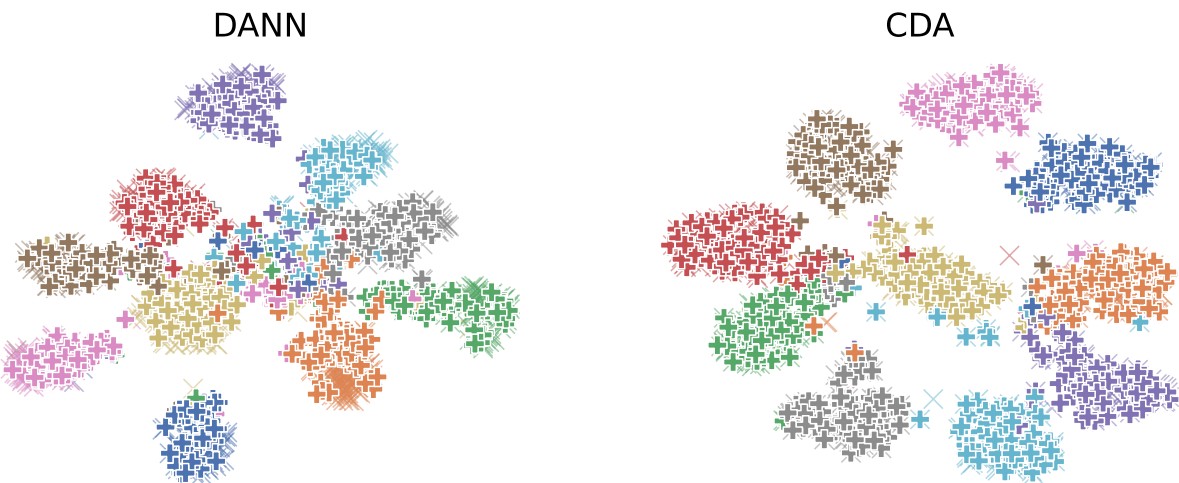

Figure 3: t-SNE visualizations for DANN and CDA to extract the contribution of the proposed contrastive module in learning domain-invariant yet class-discriminative embeddings. The analysis is for the MNIST (source) → MNIST-M (target) experiment. Each color represents one of the digits (0-9). (best viewed in color).

## 5.2 Baselines

We compare the performance of CDA with the following well-known method (a) **DANN**, which originally proposed the idea of adversarial learning for domain adaptation, and state-of-the-art methods that go beyond just domain-level alignment - (b) **MADA** and (c) **iCAN**, which use multiple domain discriminators to capture the multimode structures in the data distribution; (d) **CDAN** and (e) **CDAN+BSP**, which condition the domain discriminator on class-discriminative information obtained from the classifier; (f) **GTA**, which proposes an adversarial image generation approach to directly learn the shared feature embeddings; (g) **GVB**, which proposes a gradually vanishing bridge mechanism for adversarial-based domain adaptation; (h) **ADDA**, which uses a separate discriminative loss in addition to the adversarial loss to facilitate class-level alignment; (i) **MCD**, which uses task-specific classifiers and maximizes the discrepancy between them.

## 5.3 Implementation Details

**Network Architecture:** We use a ResNet-50 model pre-trained on ImageNet as the feature generator G. The last fully connected (FC) layer in ResNet-50 is replaced with a new FC layer to match the dimensions of the intermediate feature embedding. Both the classifier C and domain discriminator D are three-layer dense networks ($512 \rightarrow 256 \rightarrow 128$) with output dimensions of 10 (for 10 classes) and 1 (for identifying the domain), respectively.

**Training Details** The CDA network is trained using the AdamW optimizer with a batch size of 32 and 128 for the Office-31 and Digits-5 datasets, respectively. The initial learning rate is set as $5e - 4$; a learning rate scheduler is used with a step decay of 0.8 every 20 epochs. We use one NVIDIA V100 GPU for the experiments. For a detailed discussion, see the supplementary material.

## 5.4 Results

The results on the Office-31 and Digits-5 datasets are reported in Tables 1 and 2, respectively. Our proposed method outperforms several baselines across different UDA tasks. Moreover, CDA achieves the best average accuracies on both datasets. Where CDA is unable to better state-of-the-art accuracy, it reports comparable

Table 1: Classification Accuracy on Office-31 Dataset

| Method | A→D | A→W | D→A | D→W | W→A | W→D | Avg. |
|---|---|---|---|---|---|---|---|
| DANN Ganin & Lempitsky (2015) | 79.5 | 81.8 | 65.2 | 96.4 | 63.2 | 99.1 | 80.8 |
| MADA Pei et al. (2018) | 87.8 | 90.0 | 70.3 | 97.4 | 66.4 | 99.6 | 85.2 |
| iCAN Zhang et al. (2018) | 90.1 | 92.5 | 72.1 | **98.8** | 69.6 | **100** | 87.2 |
| CDAN Long et al. (2018) | 91.7 | 93.1 | 71.3 | 98.6 | 69.3 | **100** | 87.3 |
| CDAN+BSP Chen et al. (2019b) | 93.0 | 93.3 | 73.6 | 98.2 | 72.6 | **100** | 88.4 |
| GTA Sankaranarayanan et al. (2018) | 87.7 | 89.5 | 72.8 | 97.9 | 71.4 | 99.8 | 86.5 |
| GVB Cui et al. (2020) | **95.0** | **94.8** | 73.4 | 98.7 | 73.7 | **100** | 89.3 |
| CDA (ours) | 93.6 | 94.0 | **74.7** | 98.6 | **78.9** | **100** | **89.9** |

Table 2: Classification Accuracy on Digits-5 Dataset

| Method | MNIST→MNIST-M | MNIST→USPS | SVHN→MNIST |
|---|---|---|---|
| DANN Ganin & Lempitsky (2015) | 84.1 | 90.8 | 81.9 |
| ADDA Tzeng et al. (2017) | - | 89.4 | 76.0 |
| CDAN Long et al. (2018) | - | 95.6 | 89.2 |
| CDAN+BSP Chen et al. (2019b) | - | 95.0 | 92.1 |
| MCD Saito et al. (2018) | - | 96.5 | 96.2 |
| CDA (ours) | **96.6** | **97.4** | **96.8** |

[*] Best accuracy shown in **bold** and the second best as underlined.

results with the best accuracy score. A direct comparison can be made with DANN (see section 4.5), with which it shares the same adversarial component, to highlight the effectiveness of the contrastive module. On average, CDA improves the accuracy on *Office-31* and *Digits-5* by approximately 9% and 11%, respectively, compared to DANN. Furthermore, CDA significantly outperforms two well-known approaches - MADA and CDAN - that also explicitly align domains at the class level.

### 5.5   CDA Hyperparameters

The choice of hyperparameters for training the CDA model is presented in Table 3. All values are searched using the random search method. In addition to the variables presented, we use a learning rate scheduler with a step decay of 0.8 every 10 (20) epochs for training CDA on *Office-31* (*Digits-5*). We also use a dropout value of 0.2-0.5 in the second to last dense layer in the classifier $\mathcal{C}$ and domain discriminator $\mathcal{D}$.

### 5.6   Ablation Study

One of the key contributions of this work is that the proposed contrastive module can be easily embedded within existing adversarial domain adaptation methods for improved performance. We demonstrate this using the DANN architecture by Ganin et. al. as the baseline model and embed the contrastive module to improve the average classification score by 9% and 11% on the *Office-31* and *Digits-5* dataset, respectively. The training procedure (for DANN) requires minimal changes to adapt to the additional contrastive module (see Algorithm 2).

We plot the t-SNE embeddings corresponding to the last layer in the respective classifiers (of DANN and CDA) for the MNIST to MNIST-M task (Figure 3). It can be seen that the contrastive module improves the adaptation performance. For DANN, although the source and target domain align with each other, labels are not well discriminated. The reason is that the original DANN approach does not consider class-discriminative information and only aligns at the domain level. As a result, feature embeddings near the class boundaries are prone to be misclassified, resulting in lower classification accuracy on the target domain, as can be seen in the case of DANN in Table 2. For CDA, the contrastive module first increases the inter-class

Table 3: CDA Hyperparameters

| Variable | Description | Value |
|---|---|---|
| $lr$ | Learning Rate | 5e-4 |
| $bs$ | Batch Size | 32, 128* |
| $\tau$ | Temperature scaling in contrastive loss | 0.5 |
| $E$ | Total no. of training epochs | 90, 200* |
| $E'$ | No. epochs when stage-I of training ends | 25, 40* |
| $E''$ | No. of epochs when $\mathcal{L}_{CrossCL}$ is added to the overall objective function | 35, 60* |
| $\lambda$ | Scaling factor for adversarial loss $\mathcal{L}_{Adv}$ | $varies^+$ |
| $\beta$ | Scaling factor for $\mathcal{L}_{CrossCL}$ | $varies^+$ |

$^*$ The first values corresponds to the *Office-31* dataset and second value is for experiments involving *Digits-5*.
$^+$ The values increase from 0-1 using Eq. 8 and 9 to vary $E'$ and $E''$.

---

**Algorithm 2:** Contrastive Module Embedded in DANN

---

**Input**   : labeled source dataset $D_s = \{\mathcal{X}_s, \mathcal{Y}_s\}$, unlabeled target dataset $D_t = \{\mathcal{X}_t\}$, max epochs $E$, iterations per epoch $K$, model $(\mathcal{C}, \mathcal{D}, \mathcal{G})$
**Output:** trained model $(\mathcal{G}, \mathcal{C})$

**for** $e = 1$ *to* $E$ **do**
    **for** $k = 1$ *to* $K$ **do**
        Sample batch $\{x_s, y_s\}$ from $D_s$ and compute $\mathcal{L}_{SupCL} + \mathcal{L}_{CE}$
        **if** $e \geq E'$ **then**
            Sample batch $\{x_t\}$ from $D_t$ and compute $\mathcal{L}_{Adv} + \mathcal{L}_{CE}$
            **if** $e \geq E''$ **then**
                $\mathcal{L}_{SupCL} = 0$
                Generate pseudo-labels $y_t$ and compute $\mathcal{L}_{CrossCL}$
            **end**
        **end**
        Compute $\mathcal{L}_{Total}$; Backpropagate and update $\mathcal{C}, \mathcal{D}$ and $\mathcal{G}$
    **end**
**end**

Note: The additional steps for training the contrastive module are shown in blue. For DANN, $E', E'' = 0$.

---

separation in the source domain. It then aligns samples belonging to the same class across domains close to each other - leading to well-separated decision boundaries and improved classification accuracy. We conclude that with minimal tweaks to the training process, the proposed contrastive module in CDA can be embedded in existing adversarial methods for UDA for improved performance (see Figure 4 in Appendix A).

# 6 Conclusion

This paper proposes a new method for unsupervised domain adaptation (UDA) called Contrastive-adversarial Domain Adaptation (CDA). CDA improves upon existing adversarial methods for UDA by using a simple two-stage contrastive learning module that achieves well-separated class-level alignment in addition to the domain-level alignment achieved by adversarial approaches. CDA achieves this end-to-end in a single training regime unlike some of the existing approaches. Furthermore, the contrastive module is proposed as a stand-alone component that can be embedded with existing adversarial methods for UDA. Our proposed method achieves better performance than several state-of-the-art methods on two benchmark datasets, demonstrating the effectiveness of our approach. Lastly, this work further motivates an emerging research area exploring the synergy between contrastive learning and domain adaptation.

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

## A  Appendix

### A.1  CDA Hyperparameters

The choice of hyperparameters for training the CDA model are presented in Table 3. All values are searched using the random search method. In addition to the variables presented, we use a learning rate scheduler with a step decay of 0.8 every 10 (20) epochs for training CDA on *Office-31* (*Digits-5*). We also a dropout value of 0.2-0.5 in the second to last dense layer in the classifier $\mathcal{C}$ and domain discriminator $\mathcal{D}$.

Table 4: Hyperparameters

| Variable | Description | Value |
|---|---|---|
| $lr$ | Learning Rate | 5e-4 |
| $bs$ | Batch Size | 32, 128* |
| $\tau$ | Temperature scaling in contrastive loss | 0.5 |
| $E$ | Total no. of training epochs | 90, 200* |
| $E'$ | No. epochs when stage-I of training ends | 25, 40* |
| $E''$ | No. of epochs when $\mathcal{L}_{CrossCL}$ is added to the overall objective function | 35, 60* |
| $\lambda$ | Scaling factor for adverarial loss $\mathcal{L}_{Adv}$ | $varies^+$ |
| $\beta$ | Scaling factor for $\mathcal{L}_{CrossCL}$ | $varies^+$ |

$^*$ The first values corresponds to the *Office-31* dataset and second value is for experiments involving *Digits-5*.

$^+$ The values increase from 0-1 using Eq. 8 and 9 (main text) to vary $E'$ and $E''$.

### A.2  Training Process

We use the AdamW optimizer with a learning rate scheduler and a multi-step objective function for training CDA. For the first $E'$ epochs, we only optimize the supervised contrastive loss and the standard cross-entropy loss (for the classification task of interest) on the labeled source domain. The loss $\mathcal{L}$ till $E'$ is:

$$\mathcal{L}_{e<E'} = \mathcal{L}_{SupCL} + \mathcal{L}_{CE}$$

From epoch $E'$ to $E''$ the adversarial loss $\mathcal{L}_{Adv}$ is gradually added to the objective function using a ramping function $\lambda$ with value 0 at $E'$ and 1 at $E''$. Between $E'$ and $E''$, the overall loss function is:

$$\mathcal{L}_{E' \leq e < E''} = \lambda * \mathcal{L}_{Adv} + \mathcal{L}_{SupCL} + \mathcal{L}_{CE}$$

After epoch $E''$, the supervised contrastive loss $\mathcal{L}_{SupCL}$ is replaced by the cross-domain contrastive loss accompanied by a similar ramping function $\beta$ to maintain the training stability. From epoch $E''$ to the end of model training at epoch $E$, the overall loss function is:

$$\mathcal{L}_{E'' < e \leq E} = \lambda * \mathcal{L}_{Adv} + \beta * \mathcal{L}_{CrossCL} + \mathcal{L}_{CE}$$

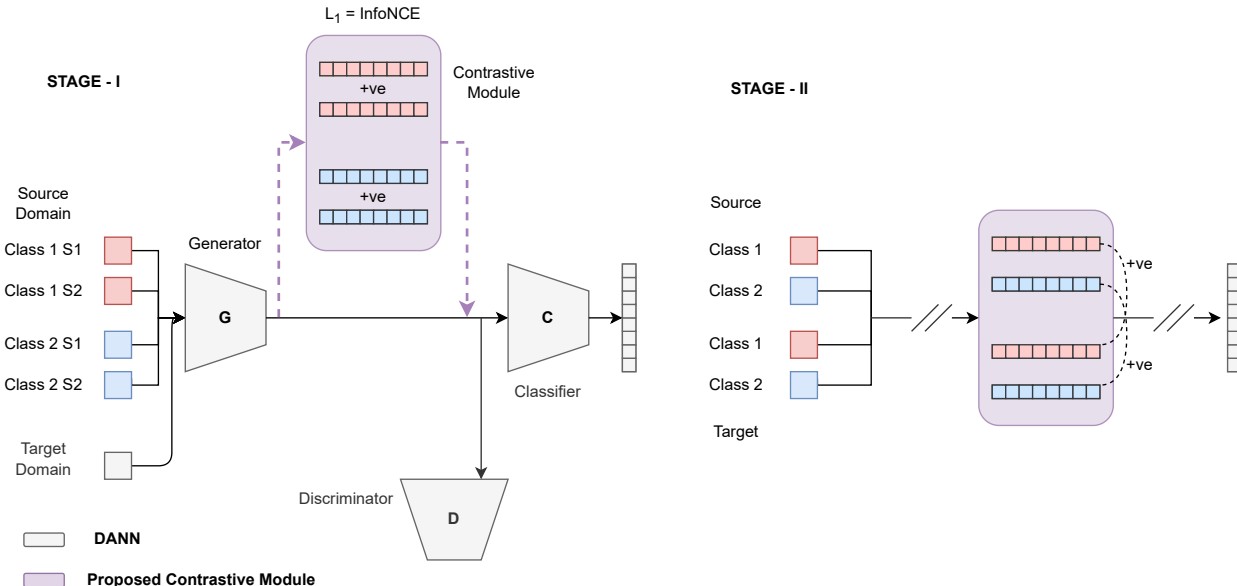

Figure 4: Illustrative diagram to demonstrate how the proposed contrastive loss module can be added to an existing domain adaptation model such as DANN (Ganin et. al). The components in gray denote parts of DANN and the contrastive module is shown in purple. Feature embeddings generated by the DANN generator pass through the contrastive module before moving to the domain discriminator and downstream task classifier. Depending on the training stage, the contrastive module minimizes either a supervised contrastive loss (stage I) using only the labeled source domain or a cross-domain contrastive loss (stage II) using both the source and target domain samples. (best viewed in color)

