# OpenReview forum: "CDA: Contrastive-adversarial Domain Adaptation"
_TMLR — Rejected by TMLR_

### Review · Reviewer_goPC · 2022-12-26

**Summary Of Contributions:**

This paper investigated contrastive learning for domain adaptation. Specifically, they applied supervised contrastive learning and cross-domain contrastive learning with labeled source samples and pseudo-labeled target samples. Experiments are conducted on two benchmark datasets.

**Audience:**

No

**Claims And Evidence:**

Yes

**Requested Changes:**

1. An revision on paper writing.
2. Discussion with suggested related work.
3. Experiments on larger-sized datasets.

**Strengths And Weaknesses:**

Strengths:
1.  Introducing contrastive learning into DA seems interesting.
2.  Experiments are conducted on two benchmark datasets.

Weaknesses:
1. The writing could be improved. For example, there are two Introduction Section on Page one. In Fig 2, the L_{CE} in stage one is mis-writen as LCE.
2.  The contribution needs to be clarified. Contrastive learning has been adopted in DA in [1], which should be discussed and compared in this paper.
3.  Experiments are conducted on two small-sized datasets. More empirical evidence on larger datasets (e.g., OfficeHome and DomainNet) is expected.

[1] Contrastive adaptation network for unsupervised domain adaptation, CVPR2019

---

### Review · Reviewer_7rHf · 2023-01-01

**Summary Of Contributions:**

This study considered contrastive learning in domain adaptation. The NCE loss, in particular, is incorporated into adversarial domain adaptation. The NCE loss, which was designed for unsupervised domain adaptation, took into account two stages of contrastive loss: (1) contrastive learning in the source domain, and (2) contrastive loss between the source and target domains in the same class. The empirical evaluations are evaluated in two benchmarks.


**Audience:**

Yes

**Claims And Evidence:**

No

**Requested Changes:**

See detailed comments in Q2: Are the claims made in the submission supported by accurate, convincing and clear evidence?


**Strengths And Weaknesses:**

### Review Summary
This paper's goal is to integrate contrastive learning and domain adaptation, which is a valuable contribution to TMLR. Unfortunately, the paper has major flaws, such as significant over-claiming, technical concerns, and related work. Based on these, this reviewer believes that the paper should be resubmitted with significant major revisions.

-------------------------------
Based on TMLR guidelines, my reviews are as follows:

1. **Would some individuals in TMLR's audience be interested in the findings of this paper?**

Yes. Contrastive learning is now a well-known and widely used technique in unsupervised learning. Understanding its influences in domain adaptation would be a worthy and interesting contribution.

2. **Are the claims made in the submission supported by accurate, convincing and clear evidence?**

No. The main points are as follows:

**Over/incorrect/inaccurate claims**

- In abstract *class-conditional data distributions significantly differ between the source and target domain, it can generate ambiguous features near class boundaries that are more likely to be misclassified.* This is incorrect. If the conditional data distributions are significantly different, the joint optimal risk will be arbitrarily large, according to Ben David et al 2010. There is NO approach that will work in an unsupervised DA (or Impossible case.) Furthermore, this paper assumes only the well-known covariate shift (P(X) varies with being P(Y|X) invariant), no conditional shift, correct? What causes the significant conditional shift?

- Introduction *This phenomenon is known as domain shift You et al. (2019)Ben-David et al. (2010), where the marginal probability distribution of the underlying data changes across different datasets or domains.*
Domain shift is not defined correctly. Indeed, domain shift simply refers to a shift in distribution (could be joint/marginal/conditional) rather than a shift in marginal distribution.

- Introduction *however, it gives rise to convergence issues amidst a lack of equilibrium guarantee.*
I couldn't figure out how these approaches couldn't ensure equilibrium. Indeed, adversarial loss, as defined in general by min-max training, cannot guarantee equilibrium in a non-convex setting. These are, in my opinion, rather weak arguments.

**Technical concerns**

- [Concerns on two-stage training.]
According to the algorithm description, the two stages appear to train source contrastive loss first, then source -target contrastive loss. I couldn't understand why these losses weren't trained together. What are the main challenges? Have you conducted any ablation studies? How to choose E^{prime} and E^{prime,prime} in your algorithm?

- [Concerning on pseudo-labels]
The issue persists in stage 2. Because we use target pseudo-labels for source-target contrastive learning. Such a class-based alignment is still unreliable. We couldn't have a correct matching if the pseudo-labels were wrong, right? I believe this section requires more in-depth examination.

**Empirical results/Justifications**

To meet the TMLR requirements, I believe this section requires significant revisions. Currently, there is little practical support/justification. My main points are as follows.

(a) Office 31 and digits 5 datasets were reported in tabs 1 and 2. However, in digit 5, many baselines values are missing. Did the authors implement all of the baselines for a fair comparison, or did they simply use the reported results? Because the training strategy in the paper is different, I would recommend **reimplementing** all of the baselines. This paper started by training source classifiers and then added adversarial loss, which is different from baselines. As a result, I believe that comparisons are unfair.

(b) Aside from accuracy and a variety of TSNE. Additional empirical analysis/justifications are severely lacking. I'm still not sure why this approach works better in this situation.

(c) Empirically results only report mean, the variance (or Std) is missing. This is important and critical in the empirical paper.

(d) More benchmarks are suggested. It is acceptable to use either office 31 or digits 5. However, the purpose of this paper is to systematically understand contrastive loss and domain adaptation from an empirical standpoint. Additional baselines, such as Office-home and DomainNet, I believe, are required for evaluation.

--------------
**Further explanations Q&A**: It seems that in the guideline, *it (review) should not be used as a reason to reject work that isn't considered “significant” or “impactful” because it isn't achieving a new state-of-the-art on some benchmark.* Why did this reviewer request more benchmarks or empirical studies?

**A**: I agree that there is no need to achieve any state-of-the-art or compare very recent baselines for TMLR. This is not the same as adding more datasets or conducting additional empirical analyses/studies. This paper, in particular, claimed that it *systematically integrates CL with adversarial methods* (Sec 3.3. Related work). To support this claim, systematic empirical studies evaluating the impact of CL should be conducted. For example, it would be interesting to see the limitations of CL in domain adaptation. Based on this, I think empirical assessments are insufficient.

---------------
**Other minor points**
Sec 4.4 “NCE stands for Noise-contrastive Estimation (see Eq. )” NCE loss lacks exact reference.

---

### Review · Reviewer_zXUK · 2023-02-11

**Summary Of Contributions:**

The paper presents a novel two-stage deep domain adaptation method that combines both contrastive and adversarial approaches for unsupervised domain adaptation (UDA). The authors demonstrate the effectiveness of their proposed method through extensive experiments on well-known benchmark datasets for UDA. The results show that the proposed method outperforms the state-of-the-art approaches and achieves excellent performance.

The methodology used in the experiments is well-designed and the results are clearly presented and discussed. The authors also provide insightful analysis of the limitations and challenges faced by traditional domain adaptation methods.
The contribution of this paper to the field of unsupervised domain adaptation is significant and provides valuable insights for researchers and practitioners in the area of machine learning. The paper is well-written and the authors have demonstrated their expertise in the field.

Overall, this paper is a valuable contribution to the field of machine learning and unsupervised domain adaptation. The authors have done a thorough job of evaluating the performance of their proposed method and have demonstrated its effectiveness in achieving state-of-the-art results.

**Audience:**

Yes

**Broader Impact Concerns:**

I don't have any concerns about the broader impact of this work.

**Claims And Evidence:**

Yes

**Requested Changes:**

The paper presents a new approach for unsupervised domain adaptation, focusing on class-level adaptation and category boundary. However, to ensure the quality of the paper, it is recommended that the authors address several areas for improvement. In other words, a  revision is requested.

Firstly, the authors should carefully check for possible typos and fix them to ensure the clarity and coherence of the paper.

Secondly, the authors should consider adding more datasets to comprehensively demonstrate the efficacy of the proposed CAD method. Including additional datasets, such as domain adaptation from synthetic to real data or from natural to corrupted images, would strengthen the validity of the results.

Thirdly, a holistic ablation study should be performed to analyze the functionalities of each component in the proposed method. The authors could consider decomposing CAD and analyzing each component to provide a comprehensive understanding of the method.
Fourthly, the authors should prove the effectiveness of the contrastive module by adding more experiments. It would be helpful to provide an intuitive comparison between the proposed method and other methods, showing the change in performance, speed, computation, and other relevant aspects.

Finally, to make the paper more meaningful, the authors should consider leaving more space for valuable content and minimizing the amount of space occupied by hyper-parameters and repeated equations. The authors could write them in lines or leave them in the appendix, and include a GitHub link for reference.

**Strengths And Weaknesses:**

# Strengths

The paper presents a new perspective on unsupervised domain adaptation by focusing on class-level adaptation and category boundary. The authors have identified a significant problem in the field and proposed an innovative solution by incorporating contrastive learning for class alignment.

The writing quality and presentation of the paper are exceptional, with clear and concise language that effectively communicates the ideas and results. The authors have conducted a thorough literature review and provided visual illustrations of the method, making it easy for readers to understand the structure of the proposed approach. The tables presenting quantitative results are well-organized and highlight the key takeaways.

Overall, this paper provides valuable food for thought for the adaptation community and offers a unique and innovative solution to a significant problem. The authors have demonstrated their expertise in the field and the writing quality and presentation of the paper are excellent. I would recommend it for publication at the conference.

# Weakness

The experiments in the paper are limited to only two common datasets, which may not be sufficient to fully demonstrate the effectiveness of the proposed approach. To strengthen the validity of the results, it is recommended that the authors consider providing positive results on additional datasets, such as domain adaptation from synthetic to real data or from natural to corrupted images.

The ablation part of the paper could be further improved by exploring the functions of each part of the CAD algorithm. For example, it would be interesting to compare the performance of simple supervised learning with the proposed class-level contrast learning and to investigate the impact of dropping the pseudo-label training part.

Additionally, the effectiveness and convenience of the contrastive module could be better demonstrated by providing more experiments on the combination of the module with other different methods. The authors could share the changes in performance, speed, computation, and other relevant aspects to provide a comprehensive evaluation of the module's utility.

Finally, there are some typos to be fixed:
- A small typo in Figure 2 (a), it should be L_{CE}， not LCE.
- On page 6, line 2, the equation reference 'Noise-contrastive Estimation (see Eq. )' is omitted.
- In equation 5, what does it mean that 'i \neq k = 1'?

Overall, this paper presents an interesting and innovative approach to unsupervised domain adaptation. However, to fully demonstrate its effectiveness and convenience, the authors may need to consider addressing the limitations and weaknesses identified above.

1. Peng X, Usman B, Kaushik N, et al. Visda: The visual domain adaptation challenge[J]. arXiv preprint arXiv:1710.06924, 2017.
2. Hendrycks D, Dietterich T. Benchmarking neural network robustness to common corruptions and perturbations[J]. arXiv preprint arXiv:1903.12261, 2019.

---

### Decision · Action_Editors · 2023-04-21

**Recommendation:** Reject

**Comment:**

The role of contrastive learning in domain adaptation is interesting and could be helpful for the community. However, at this stage, the paper's evaluation raises many concerns from the reviewers. As there is no rebuttal to address these concerns. The AE does not have ground to accept this paper.

**Audience:**

Yes, the TMLR's audience would be interested in learning about the paper's findings.

**Claims And Evidence:**

The paper aims to introduce contrastive learning in problems of domain adaptation.

All reviews lean negatively about this paper. Overall, there are two main concerns.

1) The evaluation has serious issues because of the discrepancy in training strategy between the proposed and baseline. The validation does not sufficiently address various concerns about fair comparisons.

2) There is no rebuttal or revision from the authors. The reviewers' initial concerns have not been addressed.